# The Spirit Is within Us! Ritual Practices of Latin American Pentecostals in Barcelona

Wilson Muñoz-Henríquez [1], M. Esther Fernández-Mostaza [2,*] and José Julián Soto Lara [3]

1   Faculty of Education and Humanities, Universidad de Tarapacá, Arica 1000000, Chile; wemunozh@academicos.uta.cl
2   Department of Sociology, Universitat Autònoma de Barcelona, 08193 Barcelona, Spain
3   Center for Historical Studies (CEH), Universidad Bernardo O'Higgins, Santiago 8320000, Chile; josejulian.soto@uab.cat
*   Correspondence: mariaesther.fernandez@uab.cat

**Abstract:** In recent years, Christian Pentecostalism has been one of the most successful religious movements in the province of Barcelona, while the high level of immigration from Latin America has clearly been an influential factor in its development. Yet, despite the fact that Pentecostalism has played a prominent role in religious diversification in Catalonia, it has been the subject of very few studies. This paper seeks to address this gap in research and shed light on an area of fundamental importance to the movement: its ritual dimension. Drawing on ethnographic information from the Iglesia Evangélica el Vallès (Canovelles), we describe and analyze the principal channels of communication with the sacred established by the Latin American Pentecostals. In our conclusions, we show that the communicative practices developed during worship are oriented toward manifesting the presence of the Holy Spirit, leading to the emergence of ritual condensation around this symbolic force. For this purpose, practices such as "praise", the "laying on of hands" and "speaking in tongues" establish sequential and progressive communication with the Holy Spirit throughout the process of worship, culminating in mimetic communication.

**Keywords:** Latin American Pentecostalism; communication; ritual

## 1. Introduction

The growth of Pentecostalism in the province of Barcelona over the course of the last few years has been a determining factor in the process of religious diversification in Catalonia (Griera 2010; Griera and Clot 2018). It is interesting to note that many of these Pentecostal believers can be identified by their ethnic origin; Latin Americans account for a large proportion of this religious group and are to a great extent responsible for its growth. Apparently, a large number of the Pentecostals from Latin American countries already practiced this religion in their countries of origin, and it is they who have provided the momentum for the growth of this movement. This religious phenomenon is heavily influenced by the dynamics of migration (Muñoz-Henríquez and Fernández-Mostaza 2021; Fernández-Mostaza and Muñoz-Henríquez 2018), both international and local. It is in this framework of discussion that this form of Christian religiosity is inserted (Cazarin 2017; Wilkinson and Althouse 2017). Some have even independently founded small churches to continue their evangelizing mission and thereby fulfill a religious need. In fact, given the transnational characteristics of this religious movement and the growing importance of its mission work (Robbins 2004; Anderson 2007), many pastors and missionaries have been specially sent from Latin America to create new missions and churches in Catalonia and Spain, and they have established transnational information and cooperation networks.

Despite the social relevance of this phenomenon, very few systematic studies of Latin American Pentecostalism in Catalonia or in Spain have been conducted, in line with the strong migratory process that has been experienced in Spain and specifically in Catalonia

since 2001 (Griera 2010). The principal research into this movement has focused primarily on the Church of Philadelphia within the Roma community (Cantón 2001, 2004; Gay y Blasco 2000; Llera 2005; Mena 2003, 2006, 2007, 2008, 2009; Montañés 2016), although there are also some recent publications about this movement in Catalonia (Marfà 2010) and on the subject of nascent African Pentecostal churches (Griera 2010). However, the growing Pentecostal movement of Latin American origin still fails to receive the necessary attention it warrants.

The aim of this paper is to contribute to addressing this situation. Clearly, as Griera and Clot (2018, p. 57) observe, globalization has transformed space-time coordinates, and physical, cultural, social and personal distances have been redefined. Therefore, we have focused on one of the most distinctive aspects of this movement at a global level: its ritual life. Although this aspect is of radical importance to Pentecostal life, in general it has been analyzed very little (Robbins 2009; Wilkinson and Althouse 2017). In specific terms, we will describe and analyze the principal stages of the Sunday worship sessions, highlighting the different ritual practices generated to successfully communicate with the Holy Spirit. To achieve this objective, we mainly use a theoretical perspective that explores the practices and symbols used by the different actors and entities in ritual settings. We take a pragmatic and relational perspective (Houseman 2012), closely following a long tradition of symbolism and ritual studies (Turner 1973; Leach 1989; Goffman 1991).

## 2. Methodology

A qualitative methodology has been used in this research. The one adopted was ethnography, which is understood to be a family of methods and techniques that imply a direct and sustained contact with the actors who are studied. This enabled us to obtain rich and detailed information about a phenomenon of interest and its context, as well as a subsequent in-depth analysis (Willis and Trondman 2002). This clearly agrees with the research objective, since ethnography focuses on the empirical study of social practices by which the order of interaction is generated (Flick 2002). We are to approach in a privileged way the communicative practices deployed by Pentecostals within their specific context.

The study was carried out in the Evangelical Church of Vallés, a Pentecostal community located in the municipality of Canovelles, which belongs to the Vallès Oriental region, a province of Barcelona (Spain). This town has historically been characterized by a high presence of immigrant populations, especially of Latin American population in recent years. Due to the adherence of the former, the church has grown considerably over past years.

The specific unit of analysis is Pentecostal believers of Latin American origin. The following variables were used in the selection of study participants: a legal age over 18 (ethical reasons), Pentecostal belief/practitioner, Latin American origin and regular attendance at church activities. In addition, the study focused on the worship sessions where the whole community gathered, especially at Sunday sessions, which usually last between 2:20 and 2:45 h. The main data-collection technique was participant observation (Guber 2001), which was supplemented with documentary information available mainly on websites (web pages, YouTube, Facebook). The information was collected during two periods: the first was carried out for 9 months between 2010 and 2011, while the second was carried out for 3 months between 2021 and 2022. The latter sought to contrast the information collected in the first period, especially considering the potential influence of the restrictions associated with the COVID-19 pandemic on our study phenomenon. All the information was systematized with the qualitative analysis software N-Vivo 10 and was analyzed with the content analysis technique (Piñuel 2002). The presentation of results maintains the ethnographic narrative structure of the text reproduces the three main stages in Pentecostal worship.

## 3. Results

The Iglesia Evangélica del Vallès is situated on the outskirts of Canovelles, among the industrial warehouses that surround this small town. Every Sunday, from 7 o'clock in the evening, more than 100 people meet to worship together.

### 3.1. Praise: Invocation and First Contact

The first stage of the ceremony, which lasts around 1 hour, is devoted to the practice of praise and consists of a number of canticles sung by everyone in the church. A nonbeliever arriving here to witness this for the first time would probably feel they are in a rather strange place in Barcelona: polished female choirs, cutting electric guitars, noisy drums, many voices and the clapping of a crowd are usually heard throughout the first stage of worship. The impression gained is one of a big party in progress. However, against a background of catchy rhythms in a pop, rock and even a tropical style, little by little gentle ballads also emerge, which gradually slow down the rhythm of this first part of worship. Only after some months of field work was it possible to understand the importance of this practice for initiating communication with the sacred.

Symbolically, praise may be viewed as a key, since every Sunday it serves to open the worship. It has a clear temporal significance, marking the beginning of a period (with a before and an after) in which contact with the sacred will be established. This praise communicates to those present that a period of time devoted to the worship of God has begun, as well as provides this first stage of ritual with a characteristic rhythm. It is also a key on account of the specific meaning of the communication initiated through it, since through the act of praise God is explicitly asked to "open the heavens" so that the Holy Spirit may descend. One of the acts of praise that is frequently repeated provides an example of this function.

On 28 February (2010), as was customary, worship was begun by the church's praise group and everyone joined in with their songs and dances. After some songs of praise, the young person playing the drums picked up the microphone and, before beginning the next song, announced with enthusiasm:

> Now we are going to sing a song. And we are going to ask the Holy Spirit to come to us. We already know that he is here, but we are going to ask him to come to us with all his power . . . .
>
> Hallelujah! [Some of those present shout]
>
> Do you know what happens when you begin to worship the Lord? It is said that the Lord goes silent up there and begins to listen. The Lord cares for those who are worshipping him. Let us all sing together: [praise begins]
>
> Open the gates of heaven, let it rain, let it rain [this is repeated many times].
>
> (Field note. Iglesia Evangélica del Vallès, 28 February 2010.)

These semantics show the role played by praise in this initial stage of ritual: this is an invocation, a true call that seeks to establish the first contact with God. As the young man says, the Holy Spirit is present at every session of worship, but the object is that the Spirit should manifest itself in a special, more intense and powerful way. Through this first contact, the congregation will be able to ask God to literally open the heavens and send down what for Pentecostals is the primary entity: the Holy Spirit.

However, as is usually the case in ritual communication, the explicit meaning of the words of praise sung is not enough to make these entreaties effective. In fact, the characteristic repetition of this praise does not add any new information. Yet its repetition, harnessed with musical dramatization, is necessary for it to be symbolically effective. In other words, socially it is necessary to deploy a number of body communication techniques (Mauss 1971, p. 342) to make this first stage of ritual invocation effective. Only in this way will it be possible to urge the community to participate and experience the Holy Spirit. This is why, throughout the praise, those present tend to sway their bodies from side to side, with fiery bursts of speech, their voices often cracking, keeping their eyes tightly shut,

raising their hands to heaven, dancing, etc. In short, communication needs to be dramatized (Goffman 1970, p. 54; Luhmann 2007, p. 165). In this first stage, the body technique that proves most effective is raising and moving the hands, in time with the singing and gentle dancing. In this way, shouts may be uttered for God to "open the heavens", with believers showing their dependence on him by raising their hands upward in search of this first connection with the Spirit. There is, however, a great variety of praise that fulfills this role, and in very different ways, but these examples serve to explain the purpose of this practice in the first stage of ritual. We will now return to the worship and explore the next stage.

*3.2. Sermon: The Pentecostal Tabernacle*

After the period of time devoted to praise, there is usually an interlude of a less sacred nature, when the community makes all kinds of public announcements related to the church. Then there is a quick collection before the pastor proceeds to read and interpret the word of the Bible. From a ritual perspective, these activities act as mechanisms of transition (Turner 1988), necessary preparations prior to the second stage of ritual. They anticipate the new contact that the community will establish with the sacred.

Although it is the pastor who opens this second stage of worship, the sermon is only made possible once he publicly calls on God to make himself present through the Holy Spirit and to direct and deliver the message of the word; thus, the personal power of the pastor is diminished, but at the same time what he does is sanctified. It is the manifestation of the Holy Spirit that can take effect in the sermon, refreshing the listeners and even going as far as to transform their lives. In this respect, although there are a very wide variety of themes that tend to appear in the sermons every week, one of the main, cohesive and recurring points of connection is the "presence" of the Holy Spirit.

The service that was held on Sunday, 13 June (2010) was particularly illustrative in this respect. On that occasion, the pastor stressed that in ontological terms the presence of the Holy Spirit definitively underpinned the church. He added that the conversion of those attending had only been made possible thanks to this presence. Finally, and in a very special way, he declared that one of the clearest manifestations of this presence was the generation of spiritual gifts in the community, an obvious sign of the church's vitality. Without delving now into the importance of each of these fundamental issues, we would highlight a metaphor that featured strongly in the pastor's sermon and that is key to understanding the communication practices developed in worship: the idea of the human body as a tabernacle in which the presence of the Spirit is specifically manifested.

That evening, after reading a short verse from the Bible, the pastor began his interpretation by declaring that God's people had always yearned for the presence of God, an example of which could be found in the story of Moses. The pastor recounted that this servant received specific orders from God, who asked him to build a tabernacle with wood and fabric, and to situate it in the midst of the four tribes of Israel so that God could manifest himself to his people. Moses accepted and did God's will. After commenting briefly on this passage from the Bible, the pastor speculated on the difference between the church today and the position of the people of Israel as described in the Old Testament. He observed that, formerly, God needed a tabernacle to manifest himself to his people, but after the birth, death and resurrection of Jesus Christ, the situation had changed radically. His words were as follows:

> Do you know that God has given us the Holy Spirit, whereby he is no longer with us, but now he is within us? Are you aware of the privilege of no longer needing a tabernacle in the midst of the people, because now God dwells in this tabernacle, in our bodies, and we have become the temple of the Holy Spirit? The Holy Spirit is within us, lives in us and resides in us. We have something that is much better. We have the Spirit of life! The spirit of God living within us!

> (Field note. Iglesia Evangélica del Vallès, 13 June 2010.)

With regard to this point, the words of the pastor are highly illustrative, for perhaps one of the most sublime and intense manifestations of the Holy Spirit refers to the notion of

the body of believers being a sacred tabernacle. Although the idea of the presence of God is a constant concept in the lives of Christians, an internal differentiation is added to this generic criterion of inclusion of God's people, which occurs after the death of Christ and is a guiding principle in Pentecostalism: Christ has risen and he has sent his Spirit to reside not only with his people, but above all in the body of all who believe in him and in a living way. Thus, the body of every believer becomes a genuine temple. Not only is it an instrument that is required to contact the sacred, used by different cultures and religions (De Heusch 1973; Douglas 1978; Giobellina 1985), but the body of the believer—an entity that unifies the physical, the soul and the spirit—can become a sacred symbol par excellence, a means (medium) of communication in the strong sense of the term, a hierophany of the sacred. This redefinition of the distinction between the presence and absence of the Spirit is key, since it establishes a clear difference between those who are Pentecostals and those who are not.

That day, in his sermon, the pastor explained very clearly the main distinctions and ideas by which much of the Pentecostal world view is governed, and which would subsequently have a place in the worship. However, although the whole community had participated in listening to this sermon about the Holy Spirit and its importance, this dynamic had not yet made it possible to establish more direct and intensive communication between God and believers, which was reserved for the final stage of worship.

### 3.3. The Manifestation of the Spirit

Someone attending this type of Pentecostal worship for the first time may find it difficult to discern the shift in temporality that occurs in the transition from the customary sermon to the culminating stage of worship. The personal exhortations directed toward the congregation, delivered in time with a gentle musical melody, are clear indicators of this transition.

That Sunday, 13 June (2010), the atmosphere was clearly relaxed and the pastor threw out the following question at the end of his sermon: "How many of those present here are going through a difficult time?" These words were met with complete silence, while some people raised their hand in answer. Immediately, some gentle notes from the piano could be heard, accompanying the words of the pastor. The atmosphere was transformed. We had entered another time. Against the background of the music, the pastor began to increase the volume and speed of his delivery, exhorting those present with greater intensity. The chords became louder and louder, and gradually people began to break their silence, uttering words of approval and praying more fervently.

The pastor continued to talk with great vehemence and without pausing, constantly moving from one side of the stage to the other, as he asked those present about possible suffering in their lives. As he screwed his eyes tightly shut, he declared that that very night there would be a "visitation" from the Holy Spirit, which would lead to closer, more emotionally charged and intense contact with this force. However, this visitation came with a prerequisite. The pastor continued as follows:

> This evening, I want you to stand up, or if you want to kneel, kneel or sit . . . We are going to bow down before the Lord, because the King of Kings and the Lord of Lords is here in this very place . . . How many people want to be visited by God?! [The people raise their hands more and shout: Hallelujah!] God wants to visit us now tonight, but God does not visit unless there is surrender. God will not visit unless we break with a number of things; he will not visit unless we give up what we have, vanity, false wisdom, pride, selfishness and egocentrism. There will be no visitation from God unless we relinquish all this . . . .

> (Field note. Iglesia Evangélica del Vallès, 13 June 2010).

The idea of visitation used in this stage of worship is extremely powerful. Although according to the Pentecostal conception God resides in the lives of believers, through these words and the associated atmosphere, the pastor sought to explain the otherness of the Holy Spirit, for all the ritual communication that was being channeled through this worship

at that time was aimed at facilitating the manifestation of the Spirit and this "visitation". However, this could only occur once those present yielded in their hearts and surrendered their personality. Only by surrendering and giving up their identity would they become worthy of this visit.

On hearing these words, almost the entire congregation stood up, leaving just a few people kneeling in their seats, heads lowered. The general murmur, the cheering and the weeping intensified. Gradually, there was a very public provocation of the breakdown that the pastor was calling for.

At moments such as these, it is interesting to note how the body language between the pastor and the congregation becomes a key element for measuring the effectiveness of communication in the course of ritual, regulating to a certain extent the development of this communication during worship. As a result, the pastor can see that people are gradually moving closer to a particular state in which they can communicate with the Holy Spirit because of their personal surrender. In strictly ritual terms, this breakdown is a necessary stage in which a kind of personal and spiritual purification takes place before contact with the divine. It is only once individuals and the group have experienced a certain level of emotional effervescence that the Holy Spirit is able to effectively visit them and reside in them. It is in this context that one can understand the emphasis placed by the pastor in the course of his fiery words on encouraging a state of "total surrender and breakdown" that will cut through the believer and shake them to the core.

Next, the pastor summoned those persons who had problems or who felt they had strayed from God's guidance and who wished to improve their situation. He asked them to come to the front and the whole congregation would pray for them. "Tonight we are under grace and the God of all grace is in this place," he added with fervor. Some of the congregation slowly began to rise from their seats and walk down the aisles toward the rear of the room, facing the pulpit. Once there, the pastor invited them to kneel. The believers lowered their heads, while they murmured prayers, others got down on their knees, but all raised their hands toward the heavens. Most of them were sobbing or showing physical signs of suffering, frowning, pouting with their lips or squeezing their eyes tightly shut. The pastor was also overcome by emotion: this was clear from the quavering in his voice and its intensity. He then declared that he would intercede for them one by one, announcing a new ritual practice.

First, he approached a young man who was crying as he knelt down before the pulpit. Continuing to talk through the microphone at all times, the pastor gently placed his right hand on the young man's head. His voice began to increase in intensity and he repeated his words with greater frequency, while from time to time he applied a light pressure to the young man's head. The boy's hands remained outstretched to the heavens. The pastor said:

> Tell the Lord: Here I am! Here I am! Here I am! [He says this very loudly.] Your blood cleanses me Lord, your blood cleanses me Lord. I need to return to your care, I want to stand in victory! I want to be restored! I want to be healed! Spiritually, mentally, I want my faith to be healed, I want to rebuild my faith, I want to be restored to peace, to God's love . . . father, father, father, father, father! [The whole sentence is spoken quickly.] Lord, Lord, touch me! Touch! Touch them! Touch them! Touch them! Touch them! Touch them! Touch them! Touch them! Touch them! Touch them!...
>
> (Field note. Iglesia Evangélica del Vallès, 13 June 2010.)

The boy continued to weep and began to bend over, twisting his body, while the pastor vehemently repeated the words "touch them!". Finally, the boy bent himself double. The pastor had to squat down to reach him, his hand never leaving the boy's head, and he continued to exclaim with great energy. The young man lay on his back and then flung himself forward and began to stretch out his whole body against the floor, until he was prostrate. The pastor stopped touching his head and began to gently caress his shoulder and then his back. Finally, the leader stood up and moved away from the young man to

continue walking among the congregation. The murmuring of the crowd drowned out the weeping sounds of the boy, who was calmer now, lying on the floor, where he remained for some considerable time.

An excellent example of the radical importance of body posture in communication with the Spirit can be seen in the practice of the laying on of hands (thaumaturgy) described above, mediation that is intended to cure both the physical and spiritual ailments of the believer. Once it is clear that the person has surrendered and broken down, the pastor becomes a mediator between the believer and the Spirit. The pastor is largely unaware of the particular circumstances of those who come to worship and can only read their body language; by laying his hands on the believer's head, shoulders or another part of their body, he establishes a genuine channel through which the Spirit may flow, powerfully infusing the body of the believer and curing them. In fact, the sequence of the words "touch me, touch, touch them" reflects the meaning of this action on a linguistic level. The power of God, manifested in the working of the Holy Spirit, must act first on the pastor ("touch me"), who becomes a true medium. It is only subsequently, through the laying on of the pastor's hands, that the Spirit can flow into and act on the young believer ("touch, touch them").

In this practice, there is clear expression of the performativity of Pentecostal rites on a number of different levels. First, the repetition of words that cry out for the presence of the Holy Spirit really makes it possible for this force to appear and for what is requested to take place. Repeating certain words over and over again generates redundant information, as a result of which the communicative meaning of these actions is established, thereby contributing to their ritual efficacy. On the other hand, a certain theatricality is necessary to suitably stage the operation that is taking place, in which the body and its movements play a central role in mediation. Finally, in this practice, specific roles are assigned to the participants, with a clearly defined distinction between actors and spectators, while those acting as mediators adopt a strategic position, making it possible for this sacred operation to be channeled.

However, let us return to the worship. After the stage we have just described, night was beginning to fall and the atmosphere became increasingly charged with emotion. The believers continued to sob, praying aloud, and many of them hugged each other, as the pastor spoke without respite and the music from the piano became louder. All of a sudden, a drum and an electric guitar added their rhythms, seeking to up the tempo and make the atmosphere of worship more intense.

The pastor declared that the prophetic word that evening was clear: "Call to me, and I will answer you," so they had to seek the "presence" of the Lord, calling on him to intercede. Speaking with greater force, he said they had to understand that "the Spirit is power," and that the Book of Acts talks about the Spirit and its manifestations. He proceeded to open his Bible and, with great emotion, he began to read:

> When the day of Pentecost came, they were all together in one place. Suddenly a sound like the blowing of a violent wind came from heaven and filled the whole house where they were sitting. They saw what seemed to be tongues of fire that separated and came to rest on each of them. [The pastor omitted part of the verse.] All of them were filled with the Holy Spirit and began to speak in other tongues as the Spirit enabled them.

> (Field note. Iglesia Evangélica del Vallès, 13 June 2010.)

As the pastor read, the voices of the congregation rose in volume, uttering various cries of "Hallelujah!" and "Glory to God!" in addition to prayers. Just as the pastor finished these words from the Bible, a woman of about 50 who usually sang very loud and danced a lot during the worship began to speak in tongues. With her eyes closed and her hands raised, she slowly twisted her body from left to right, almost without moving her feet, and began to emit a sound like the sound made when the tongue is rapidly moved up and down inside the mouth. She was speaking in tongues. She continued for some time. Then she knelt down, sobbing inconsolably, as she continued to speak in tongues. After

about 5 min, she fell silent, prostrate on the floor, while her daughter, who had been at her side at all times, slowly caressed her back and gently took one of her hands. Meanwhile, the pastor began to intersperse his fiery interjections with lengthy phrases in tongues, which were somewhat different from the woman's earlier exclamations: "Oh shirabadshai, rabadshu. Oh yes Lord, oh yes Lord." His words were echoed by one or two members of the congregation.

The passage from the Bible read by the pastor is of relevance on account of its content, but also because of what it contributes in this context. On the one hand, it is the first reference to be found in the Bible of the manifestation of the Holy Spirit after the death of Jesus. This phenomenon and its various manifestations provide the central theme of the Book of Acts. Furthermore, the words read are drawn from the verses that clearly recount the presence of the Spirit in the form of strange tongues (glossolalia), a distinctive practice of Pentecostalism. In this respect, if we consider the radical importance of the figure of the Holy Spirit and its manifestation in Pentecostalism, the public reading of this significant passage from the New Testament serves to remind those present of the legendary origin of the movement.

On the other hand, if we bear in mind that most Pentecostals are more than familiar with this extract from the Bible, the question has to be asked: What is the purpose of reading this text? What information does it provide? While it fulfils the need to recall the origin of the movement and to demonstrate the veracity of these words, above all it shows that this seminal occurrence may be witnessed again in this particular context, having the effect of stirring up the community. Indeed, the pastor gave a public reading of this passage at a moment when the act of worship had begun to approach its climax, with a high degree of collective effervescence, and when the Holy Spirit had already acted on some of the believers before the eyes of the congregation.

## 4. Discussion

It is important to add that in the course of the sermons in the period of 2021–2022, thanks to new technologies, highly significant changes have been introduced to the "staging" of the liturgy, especially in comparison with Catholic rites. It is now common to use a spotlight to project the words of the songs and to add emphasis to the sermon with photographs; Facebook serves as a fluid channel of communication, generating a cohesion that goes beyond the walls of the church.

According to literature in the fields of both theology and social sciences, glossolalia (speaking in tongues) is one of the distinctive experiences and practices of Pentecostalism (Garma 2000; Anderson 2007; Fancello 2009). However, what it is important to underline here is the meaning and function of this communicative practice in ritual terms. It is not just any relationship with the sacred that is established through glossolalia, but rather an extreme intensification of this type of communication. Perhaps this is why it is one of the practices that most clearly shows Pentecostalism offering believers the opportunity to establish a direct connection with the Spirit. Mimetic communication is evident in the practice of speaking in tongues, where there is a rapid and intense flow of an infinite number of unintelligible vocal sounds from the believer's tongue: a celestial language is spoken, imitating the language of the divine. What is of sociological relevance here is that this is a ritual practice in which the distance between the believer and God is partially removed. This type of communication becomes so intense that the Spirit itself is understood to be putting the words into the mouth of the pastor and the believers, and even they themselves cannot understand what is being said.

As a result of this intimate and extremely intense relationship with the Spirit, the latter can take possession of the believer, residing in them and speaking angelic tongues through them. Obviously, this type of positive possession or "adorcism" (De Heusch 1973, p. 258) cannot simply be understood as discourse, for it is a practice requiring a number of body techniques that provide reliable evidence of this type of communication in a ritualized manner: hands are raised, energetic movements are necessary, changes

in tones of voice, the tongue swiftly moves to and fro, eyes are tightly closed, etc. The body in motion—arms, hands, head, tongue and manifest emotion—becomes the clear symbol of sacred happenings. It is thanks to the fact that glossolalia actually takes place, facilitated by a context of ritual, that all will understand it is not only a sublime form of communication with the divine, but communication that is also divine in itself, in which the Spirit loquaciously demonstrates its incarnation. Although the Holy Spirit resides in the praise of God's people and is present *among* them, its thunderous presence *in* the body itself of the believer is the most sublime form of communion and communication with the Holy Spirit. Without the need for direct intervention from a leader, but with the help of a ritualized context, any believer can become a genuine mediator, a medium empowered to cross the frontier between the transcendent and the immanent, embodying this limit him- or herself and without there being a need for the presence of intermediaries.

The findings and their implications should be discussed in the broadest context possible. Future research directions may also be highlighted.

## 5. Conclusions

In formal terms, the ritual practices conducted in the course of Pentecostal worship may be interpreted as acts of communication structured around the distinction between presence and absence, characteristic of the systems of co-presential interaction, in which the communication takes as a reference the information that is present in the situation (Goffman 1991, p. 173; Luhmann 2006, p. 645). This generic distinction is redefined in the case of Pentecostal worship, since communication in worship largely revolves around the distinction between the presence and the absence of the Holy Spirit, which is constantly redefined and reintroduced into other secondary distinctions in the course of ritual.

References to the presence of the Holy Spirit are so frequent that it creates a situation in which this information becomes increasingly redundant, something that is characteristic of many systems of ritual communication (Leach 1989, pp. 51–52). As worship progresses, ritual condensation is generated around the Holy Spirit, operating as a dominant symbol (Turner 1973). These symbols have the particular feature of becoming a factor of social action in ritual settings. They are the center of the group's interaction and the community conducts its acts of worship around them. Nevertheless, their most important characteristic is that they can be expressed almost directly in these contexts, triggering the manifestation of emotion among the participants and saturating the overall situation. In turn, the communication of emotions encourages participation in a ritual performance once again, as the sequence of worship shows.

We would like to complement this general conclusion with a more specific synthesis regarding the role played by the various practices that unfold in the different stages of ritual. We have shown that the principal communicative practices employed in Pentecostal worship are praise (first stage), the sermon (second stage) and thaumaturgy and glossolalia (third stage). Following the sequence of ritual, its stages gradually bring worshippers closer and closer to God and with increasing intensity, and the communicative techniques developed are directed to this effect. In the first stage, the praise constitutes an invocation to God, who is asked to open the heavens so that the Holy Spirit may descend and visit the believers. This is the first call. In the second stage, the discursive semantics of the sermon mainly refer to the properties possessed by the Spirit and the functions fulfilled by its *presence*, making the human body a temple. Finally, it is only in the last stage of ritual that the event which the community ultimately seeks takes place: the *visitation* of the Spirit, its action and incarnation in people.

Regarding the mediators, we observe, on the one hand, that the Spirit acts on the believer who is suffering through the pastor (the medium), who can gain access to the Spirit's power and channel its energy toward the believer and shows the pastor's relevance (Cazarin and Cossa 2017). On the other hand, after the moment at the root of the Pentecost episode has been recalled, the Spirit resoundingly descends, making contact with believers and even residing in them. At the same time, it generates a kind of positive possession

(adorcism), expressed in glossolalia, where the presence of the Spirit is manifested (Garma 2000; Mena 2003). In this sequence, but above all in the final stage of the ritual, it is possible to see the importance of the use of body posture appropriately staging and dramatizing this form of ritual communication, to the point of turning the believer into someone possessed, whereby this person represents the limit between the transcendent and the immanent. Body mediation is a very present phenomenon in contexts where migration and Pentecostalism are expressed together (Cazarin 2017; Wilkinson and Althouse 2017). This feature updates one of the unique resources of classical Pentecostalism (Anderson 2007).

This discussion allows us to account for the specificity of the ritual practices of probably the most important Christian religious movements currently in Barcelona, and also invites us to reflect on how this particular ritual dimension contributes to the understanding of the great expansion within the migrant population of Latin American origin in Spain.

**Author Contributions:** Conceptualization, Methodology, Investigation, Writing—Original Draft Preparation: W.M.-H. Writing—Review & Editing, Supervision, Project Administration: M.E.F.-M. Writing—Review: J.J.S.L. All authors have read and agreed to the published version of the manuscript.

**Funding:** Wilson Muñoz-Henríquez obtained funding by the UNESCO/Japan Young Researchers' Fellowship Programme, UNESCO—France and the Government of Japan, and the scholarship program *Becas Chile* N° 72150364, ANID-Chile.

**Institutional Review Board Statement:** Not applicable.

**Informed Consent Statement:** Informed consent was obtained from all subjects involved in the study.

**Data Availability Statement:** Not applicable.

**Acknowledgments:** The authors wish to thank the anonymous reviewers for their valuable comments and suggestions.

**Conflicts of Interest:** The authors declare no conflict of interest.

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
