# Peer review of "The Spirit Is within Us! Ritual Practices of Latin American Pentecostals in Barcelona"

_religions, doi:10.3390/rel13060501_

Round 1

Reviewer 1 Report

It is advisable to develop the analysis within a specific theoretical framework, considering it within a larger social phenomenon such as, for example, international migration could be. It is advisable to consult the international bibliography on religiosity experienced within a transnational context.

Author Response

We thank the reviewer for their comments and suggestions.

  1. We have incorporated a specific theoretical perspective for our analysis: Houseman 2012
  2. We have highlighted the link between our case study and migration as a broader social phenomenon (Muñoz-Henríquez and Fenández-Mostaza 2021; Fernández-Mostaza and Muñoz-Henríquez 2018; and Griera 2010)
  3. We have highlighted the Latin American migratory context in the introduction, as well as in the methodological section and the final discussion. All of them aspects that undoubtedly have improved our article

All changes are highlighted in blue.

Reviewer 2 Report

Overall, this is potentially a very good paper that offers insight into the role of Latin American Pentecostals in Spain. It generally deals with the characteristics around Pentecostal worship. However, it can be strengthened with some analysis that engages two important volumes. The first is Interactional Ritual Chains by Randall Collins, which is one of the most important books on ritual, emotion, and the body. The second is the edited volume Pentecostals and the Body by Wilkinson and Althouse (Brill, 2017) that applies much of the work of Collins on Pentecostals. More specifically, the article "Emotions and Spiritual Knowledge: Navigating (In)Stabilities in Migrant Initiated Churches" by Rafael Cazarin deals with migrant churches in Spain and may offers some insight for the authors. There is some minor editing to be done but overall, the description is very good. Some more context or details that place this congregation within the larger landscape of religion in Spain and more specifically, Pentecostals in Spain could be helpful (see "Spain" in Brill's Encyclopedia of Global Pentecostalism).

Author Response

The authors thank the reviewer for their valuable comments and suggestions.

  1. We have incorporated bibliography on rituality, body and Pentecostalism, where the perspective of Collins is clearly present (Brill 2017; Cazarin 2017).
  2. We have incorporated background information and details about our case study, in the Spanish context, especially in the Methodology section, which improves the general understanding of the article.
  3. Finally, we have incorporated the editor's suggestion adding a detailed Analysis methods and Techniques

All changes are highlighted in blue.

Round 2

Reviewer 1 Report

I suggest improving the conclusions and supporting them with references in secondary literature.

Author Response

We are glad to send you here attached our manuscript ID: religions-1698897, entitled “The spirit is within us! Ritual practices of Latin American Pentecostals in Barcelona”, after the second revision.

All changes are highlighted in green.

We remain at the disposal of the editor and reviewers for any further changes or suggestions you may have and thank your time and dedication.

Best regards,

The Authors

Reviewer 2 Report

The changes are helpful. I have one question about the reference to Brill in the text that is not in the Reference section. Also, note, that Brill is the publisher, not the author. Once the reference is fixed, I think it is fine to publish.

Author Response

(The authors gave the same response as above.)
